# Effectiveness and Protection Duration of Anti-COVID-19 Vaccinations among Healthcare Personnel in Cluj-Napoca, Romania

**DOI:** 10.3390/vaccines11030521

**Published:** 2023-02-23

**Authors:** Maria I. Brumboiu, Edina Iuga, Andreea Ivanciuc, Sergiu Mutaffof, Alice S. Tudosa, Cristina Gherasimovici, Irina Iaru

**Affiliations:** 1Epidemiology Department, Faculty of Medicine, Iuliu Hatieganu University of Medicine and Pharmacy, 400012 Cluj-Napoca, Romania; 2Hospital Epidemiology Department, University Infectious Diseases Hospital, 400348 Cluj-Napoca, Romania; 3Cluj Unit, French-Speaking International Clinical Epidemiology Network, Iuliu Hatieganu University of Medicine and Pharmacy, 400012 Cluj-Napoca, Romania; 4Pharmacology, Physiology, Pathophysiology Department, Faculty of Pharmacy, Iuliu Hatieganu University of Medicine and Pharmacy, 400012 Cluj-Napoca, Romania

**Keywords:** vaccine effectiveness, COVID-19, healthcare personnel, protection duration

## Abstract

The anti-COVID-19 vaccines, developed for use during the pandemic period, must be evaluated for effectiveness in order to coordinate the vaccination program. Therefore, this study aimed to measure the anti-COVID-19 vaccine effectiveness (VE) and duration of protection against symptomatic forms of infection among healthcare personnel who were professionally exposed to the SARS-CoV-2 virus. A prospective cohort study, which was conducted in a university hospital between January 2021 and April 2022, compared immunologically naïve and previously infected personnel who were vaccinated, revaccinated, or unvaccinated. The VE was measured based on survival rates constructed with the actuarial method, using 30 day intervals. Among the 783 subjects that were included in the study, those that were vaccinated showed a decrease in VE from 90.98% (95% confidence intervals (CI): 74.87–96.77) in the first 30 days to 69.95% (95% CI: 40.29–84.87) at 60 days after vaccination. The VE for revaccinated personnel was 93.27% (95% CI: 77.53–97.99) at 60 days and 86.54% (95% CI: 75.59–92.58) at 90 days after revaccination. For previously infected personnel, protection against reinfection was 94.03% (95% CI: 79.41–98.27) at 420 days and 82.08% (95% CI: 53.93–93.03) at 450 days after revaccination. The highest VE for preventing the symptomatic forms of COVID-19 was observed in the revaccinated, but only for a 3-month duration. Better protection against reinfection was provided by revaccination after passing through infection.

## 1. Introduction

The anti-COVID-19 vaccine was required to overcome the COVID-19 pandemic (declared by the World Health Organization (WHO) on 11 March 2020) and the morbidity and mortality it caused, as well as the massive impact of the non-pharmacological measures on society’s daily life [1,2,3,4]. Producing an effective vaccine for the SARS-CoV-2 virus required remarkable effort, not only to identify the antigens inducing the protective immune response, but also to make it available in the shortest time possible [2,5].

The first anti-COVID-19 vaccine preparations that were conditionally approved by the European Medicines Agency for use during the pandemic were Comirnaty (BioNTech and Pfizer) in December 2020, Spikevax (Moderna), and Vaxzevria (AstraZeneca) in January 2021, and COVID-19 Janssen in March 2021 [6]. Their efficacies, evaluated in clinical trials, were 95% (95% confidence intervals (CI): 90–97.9) for Comirnaty, 94.1% (95% CI: 89.3–96.8) for Spikevax, 74% (95% CI: 65.3–80.5) for Vaxzevria, and 66.9% (95% CI: 59–73.4) for COVID-19 Janssen [6,7]. 

Compared to clinical trials, vaccine administration programs in real-world populations may have significantly different efficacies [8,9]. Evaluating the effectiveness of real-world practices may identify factors that significantly affect vaccine efficacy, including aspects that were not previously noticed in clinical trials. Therefore, the collection of field data related to vaccine effectiveness (VE) and post-vaccination protection is necessary for documenting and supporting the coordination of the national vaccination program.

In addition to the quality of the vaccine preparations, VE is also influenced by the immune response of the vaccinated person, the conditions under which the vaccination programs are carried out, and the risk of exposure to the etiological agent [8,10,11]. In order to properly measure VE, epidemiological studies must have the capacity to control the influence of these factors. In the study of VE, prospective cohort studies are recommended when the incidence is high enough to reflect homogeneous exposure to the etiological agent, the vaccine data are available and correctly recorded, both vaccinated and unvaccinated people exist in the population, and the cohorts can be actively surveilled [8,11,12]. These criteria were met by the COVID-19 pandemic, but it is neither time- nor cost-efficient to conduct the study in the general population. Amongst population categories that are at risk of acquiring the infection, healthcare personnel were the first to be vaccinated at the beginning of the pandemic [11]. At the same time, the information that is obtained from this professional category is usually of high quality.

Therefore, the objective of this study is to measure the effectiveness of anti-COVID-19 vaccination and duration of protection against symptomatic forms of infection among healthcare personnel professionally exposed to the SARS-CoV-2 virus.

## 2. Materials and Methods

### 2.1. Study Design

A prospective cohort study was conducted among healthcare staff working in a university hospital where patients with COVID-19 were treated. The comparison groups were the vaccinated and non-vaccinated cohorts, dynamically constituted depending on the implementation of the vaccination program and the acceptance of the anti-COVID-19 vaccination by the personnel (Figure 1).

Vaccination was initiated on 27 December 2020, with the Comirnaty vaccine, and revaccination began on 28 September 2021. Considering the onset of post-vaccination protection ≥ 14 days after the vaccine dose administration, partially immunized personnel began to accumulate from 10 January 2021, and were protected by revaccination from 11 October 2021. A particular group consisted of those being immunized by passing through the disease before the vaccine introduction, without being vaccinated afterwards. Their identification was possible through the active surveillance that was implemented as a preventive measure for hospital staff on 27 February 2020, when the first suspected cases of SARS-CoV-2 infection were admitted to the hospital. This was a continous process until 30 April 2022, which was shortly after lifting the state of alert (8 March 2022) and the majority of non-pharmacological measures that were implemented due to the COVID-19 pandemic. As a result, the cohort inclusion period for measuring VE was 10 January 2021–30 April 2022.

Active staff surveillance for COVID-19 consisted of daily clinical triage upon entry to shift, for fever and acute respiratory symptoms that were suggestive of COVID-19, and laboratory testing with real-time reverse transcription polymerase chain reaction (rRT-PCR), every 2 weeks. After the introduction of vaccination, testing was less often and irregular, mostly for personal interest in various situations (international travelling or participation in socio-cultural or sportive events, etc.) for which a negative COVID-19 test was required. For all cases that were confirmed by rRT-PCR, an epidemiological investigation was carried out on the day of confirmation or within a maximum of the first 3 days, by a team of three epidemiologists.

### 2.2. Participants

The study sample consisted of the permanently employed healthcare staff of the hospital, from all wards, who actively worked for more than 6 months, between 27 February 2020 and until 30 April 2022.

The criteria that was used to establish the comparison groups was the presence of vaccination and the subsequent diagnosis of a symptomatic episode of COVID-19. The diagnosis of COVID-19 and the classification of the clinical forms were both made according to the WHO methodology [13]. Asymptomatic cases with a positive rRT-PCR were excluded.

The occupational risk of exposure to SARS-CoV-2 for staff was controlled by the use of appropriate personal protective equipment throughout the work shifts and active surveillance. The community risk was analyzed through the publicly available monthly incidence report at the national level using the reported cases to the European Centre for Disease Prevention and Control as the data source [4].

### 2.3. Data Sources and Variables

The data sources for the COVID-19 cases were the reports of the epidemiological investigation that were similarly conducted for each confirmed case among the employed staff. The data that were recorded were the presence of symptoms, onset date, confirmation by rRT-PCR (date of the positive result), date of birth, gender, professional category, hospital department, probable source of infection, direct contact with other employees, clinical form of COVID-19, concomitant pathology, hospitalizations, death, previous episode(s) of COVID-19, and vaccination data (preparation, trade name, doses, and administration date). Vaccination data for all persons that were included in the study were documented through the Vaccination Register or the certificate of vaccination. The data for the personnel who did not have COVID-19 were extracted from the active surveillance electronic register that is used for personnel monitoring during the pandemic state of emergency and alert.

The fully (or completely) vaccinated persons were considered those who had had two administered doses of the Comirnaty, Spikevax, or Vaxzevria vaccines or one dose of the Janssen vaccine. The revaccinated persons (or those having received a booster) were those with a dose that was administered more than 6 months after complete vaccination. The breakthrough infection was the episode that occurred more than 14 days after the complete vaccination or after the revaccination dose. Those persons who had the onset of the disease less than 14 days after the last vaccine dose were classified according to the status of the earlier dose. Those persons who had a partial vaccination schedule were considered unvaccinated. Those persons becoming immune through a symptomatic episode before the introduction of the vaccination, and who were not vaccinated later, formed the naturally protected cohort (or the left-censored group).

The data were anonymously recorded in an electronic database that was developed with the Excel computer program. To avoid recording errors and to fill in missing data, the electronically registered data were manually checked twice by confronting them with the primary documents of the epidemiologically investigated cases. Each person’s vaccination status and the date of disease onset were carefully watched for correct classification into comparison groups. The verification of data that raised a suspicion of error during the primary descriptive data analysis was repeated.

Only the investigation team had access to primary data, ensuring confidentiality. Epidemiological investigations were possible on the legal basis provided by ministerial orders 555/7.04.2020 and 1513/9.09.2020 and the law 55/18.05.2020, with the positive approval of the ethics committee for conducting the research (AVZ45/18.02.2022).

### 2.4. Statistical Methods

The quantitative data were analyzed by indicators of central tendency and dispersion. The statistical significance of the difference between the means of variables not following a normal distribution was tested with the Kruskal–Wallis test.

The qualitative data describing the main characteristics of the subjects were analyzed by frequency distributions in contingency tables, and statistical significance was tested with the Fisher or Chi-square test.

VE was calculated as:VE = [(1 − relative risk) × 100].(1)

Relative risk (RR) was obtained based on survival rates that were constructed with the actuarial method, for 30-day intervals. The RR was the ratio between the risk in vaccinated and non-vaccinated persons with the 95% CI being obtained through:95% CI = exp[logeRR ± 1.96 × SE],(2)
where exp is the exponential, log_e_ is the natural logarithm and SE stands for standard error of the log RR [14].

Protection for the second symptomatic episode in the left-censored individuals was analyzed through the RR reduction (RRR) compared to the unvaccinated who had the first episode in the period after the introduction of vaccination.

The threshold value for statistical significance was 0.05, and statistical processing was performed with the Addinsoft Xlstat statistical program.

## 3. Results

A total of 783 people who were professionally active in the period January 2021–April 2022 were included in the study. Their main characteristics were an average age (in the middle of the surveillance period) of 45.06 years (range 19.65–68.76 years), the predominance of females (81.86%), the nurses as a professional category (37.16%), and the work activities that were carried out in the clinical wards (48.15%) (Table 1).

The cohorts for measuring the effectiveness of the anti-COVID-19 vaccination during the surveillance period were the vaccinated (241 people, and after reclassification another 3 cases were added); revaccinated (222 people, of which 3 were reclassified to the vaccinated when they got COVID-19); unvaccinated (170 people), vaccinated (67 people) or revaccinated (40 people) after the first episode of COVID-19; and the naturally immunized cohort (43 people). A total of 335 symptomatic cases were identified, of which 40 had a second symptomatic episode. The excluded cases were 44 asymptomatic episodes of personnel from all departments (10 administrative staff, 13 care staff, 18 nurses, and 3 physicians).

Between those who had had and those who had not had COVID-19, the average age of the individuals was considered significantly (*p* = 0.054) lower (41.85 years) in those with a second symptomatic episode. For the professional category, significant differences (*p* = 0.010) resulted from the higher share of nurses being affected by COVID-19 both as a first episode (43.28%) and as a second (55%). According to the unit where they worked, those from the clinical departments, a significantly (*p* = 0.025) lower proportion (44.18%, compared to 35%), were affected by COVID-19 during the surveillance period. Regarding the anti-COVID-19 vaccination, a significantly higher proportion of unvaccinated (42.5%; *p* = 0.046) were in the group of those who had two episodes, compared to the other groups. Among all COVID-19 cases, 138 (41.19%) were breakthrough infections, of which 80 (23.88%) were in fully vaccinated persons and 58 (17.31%) were in revaccinated persons.

The most common vaccine preparation was the Comirnaty vaccine, without significant differences between the groups. Among the non-vaccinated, three people received one shot of Comirnaty, after which the vaccination was interrupted due to the side effects, such as anaphylaxis in two cases (without COVID-19 during the surveillance period) and the reactivation of autoimmune thyroiditis for the third (with COVID-19 during the surveillance period).

The cases in the second symptomatic episode compared to those in the first were more frequently mild forms of disease (92.5% versus 81.79%; *p* < 0.0001), with none being severe or requiring hospitalization (Table 2). At the same time, they needed medical attention less often (27.50% versus 57.61%; *p* < 0.001). There were no significant differences regarding the presence of concomitant pathology or the source of contamination.

The risk of community exposure to the SARS-CoV-2 virus, assessed by monthly incidence at the national level, remained high with variations depending on the pandemic waves (Figure 2). The cases among healthcare personnel evolved parallel to the community waves except in 2021, when there were persons that were protected by vaccination and then by revaccination.

The VE to prevent a first symptomatic episode of COVID-19 was higher in the revaccinated cohort, with a value of 93.27% (95% CI: 77.53–97.99) at 60 days and of 86.54% (95% CI: 75.59–92.58) at 90 days after revaccination, but with a rapid decrease at 120 days, then reaching negative values (Figure 3). For the fully vaccinated cohort, protection decreased from 90.98% (95% CI: 74.87–96.77) for the first 30 days to 69.95% (95% CI: 40.29–84.87) at 60 days, with the decline continuing to negative values 270 days after full vaccination. A slight increase was observed at 120 days coincided, according to the monthly monitoring, with a low community exposure to the SARS-CoV-2 virus, but thereafter, the decrease in VE continues until the end of the surveillance period.

Protection for the second symptomatic episode of COVID-19 was higher in the cohort of revaccinated persons after passing through the infection, with an effectiveness of 94.03% (95% CI: 79.41–98.27) at 420 days and 82.08% (95% CI: 53.93–93.03) at 450 days (Figure 4). This level of protection was close to the left-censored ones. They were protected by natural immunization during the first 360 days, and the RRR started to decrease from 390 days (93.18%; 95% CI: 70.15–98.44) with a slow decline, reaching a level of 88.88% (95% CI: 71.33–95.69) at 420 days. The effectiveness for the revaccinated group could not be calculated, having only one case in this category.

## 4. Discussion

The present study aimed to evaluate the effectiveness of vaccination against COVID-19 and met the main conditions required for its use, i.e., the pandemic period conferring homogeneous exposure of the subjects, the presence of both vaccinated and non-vaccinated persons among the healthcare personnel depending on the vaccination acceptance, and accurate data records through active surveillance that were constantly applied by the same team of doctors [8,11,12,15]. Meeting these criteria contributed to avoiding the main biases, which could have affected the accuracy of the study [8,11,12]. Other possible sources of bias could have been from undiagnosed COVID-19 cases due to confusion and self-medication when symptomatology might have overlapped with chronic conditions, or from persons who were vaccinated and did not present the vaccination document. However, these events were likely rare due to the active surveillance and verification of data using several sources.

Among the healthcare personnel, the predominance of females had no influence on the VE; the studies that have bveen carried out so far have shown that the difference between the sexes regarding the immune response to the COVID-19 vaccines were statistically insignificant [16].

The VE that was identified in the present study was close to the 95% efficacy (95% CI: 90–97.9) recorded in clinical trials for the Comirnaty vaccine, which was the most common vaccine here [6]. Similar results were obtained for the Comirnaty vaccine in several field studies elsewhere. The cohort study that was carried out on medical staff in England, between 7 December 2020 and 5 February 2021, found a VE of 85% (95% CI: 74–96) for the prevention of symptomatic and asymptomatic forms in the first 7–14 days after full vaccination [17]. Similarly, the study on medical personnel in Treviso province, Italy, between 27 December 2020 and 24 March 2021, found an effectiveness of 94% (95% CI: 51–99) for the prevention of symptomatic forms at 7 days from the second vaccine dose [18]. In a test-negative case–control study that was conducted with U.S. healthcare workers, an effectiveness of 88.8% (95% CI; 84.6–91.8) against symptomatic forms was found during the period 28 December 2020–19 May 2021 [19]. In another study in the general population that was conducted in Israel, from 20 December 2020 to 1 February 2021, the effectiveness for the prevention of symptomatic forms was 94% (95% CI: 87–98), 7 days after the second vaccine dose [20].

The duration of high effectiveness that was observed in the present study was only 2 months for the vaccinated and 3 months for the revaccinated. Similarly, in the groups that were included in the clinical trials evaluating the efficacy of the Comirnaty vaccine, an effectiveness of 83.7% (95% CI: 74.7 to 89.9) was found 4 months after dose 2, with an average rate of decline of about 6% every 2 months [21]. However, a case–control study that was conducted in Qatar to estimate the duration of protection of vaccination with the same mRNA vaccine against symptomatic forms of infections with the BA.1 and BA.2 variants, found an effetiveness level of 46.6% (95% CI: 33.4–57.2) and 51.7% (95% CI: 43.2–58.9) in the first 3 months after dose 2, respectively, and 59.9% (95% CI: 51.2–67) and 43.7 (95% CI: 36.5–50) in the first month after revaccination [22]. An observational population study in the Capital Region of Denmark, during July–September 2021, found a hazard ratio of a positive RT-PCR test for vaccinated to unvaccinated of 0.2 (95% CI: 0.05–0.48; *p* = 0.001) for infection in 8 months of follow-up after the second dose of Comirnaty [23]. In a systematic review and meta-regression that was carried out with data from randomized clinical trials and observational studies, published between June and December 2021, the average change in vaccine efficacy or effectiveness was estimated at 24.9% (95% CI: 13.4–41.6), from 1 to 6 months after complete vaccination with Comirnaty, Spikevax, Janssen, or Vaxzevria, for symptomatic forms, at all ages [24].

These results certify the fact that vaccination has high effectiveness, but this wanes over time through the loss of immune protection and the dynamics of the SARS-CoV-2 virus antigenic changes (the circulating variants of SARS-CoV-2). The predominant variants circulating nationally in the last months of 2020 were the ancestral strain, the Alpha strain in the first 6 months of 2021, then Delta in the second half, followed by Omicron in the first months of 2022 [25]. In the present study, the circulating variants were not followed because genomic sequencing was not performed on staff cases.

Considering the natural protection, we noticed its reduced efficacy due to the evidence that of those who passed through the infection, some had a second symptomatic episode of COVID-19 at an interval of just over a year. Furthermore, among those with two COVID-19 episodes, the proportion of these cases was significantly higher than in the other categories. Similar to the vaccination, this also suggests that the natural decline of protection or the circulation of antigenically different variants caused no remaining protection from the previous exposure.

Among the published studies, in a test-negative case–control study in the general population of Qatar, the effectiveness for preventing symptomatic reinfection with BA.4 and BA.5 subvariants was 35.5% (95% CI: 12.1–52.7) in the case of previous infection in the pre-Omicron period and 76.2% (95% CI: 66.4–83.1) for the first infection in the post-omicron period [26]. In the same population, another national cohort study compared the period of dominance of the original circulating variant of SARS-CoV-2 from the first wave with the period of the second and third waves, when the B.1.1.7 (alpha) and B.1.351 (beta) variants dominated; re-infections were rare, and the clinical forms of evolution were milder, with the odds of severe disease at 0.12 (95% CI: 0.03–0.31) for reinfection with the circulating variant [27]. In a retrospective observational study that was carried out between January and February 2021 (Delta variant dominated) in Qatar, those that were vaccinated with two doses of Comirnaty had a 13.06 times (95% CI: 8.08–21.11) higher risk of breakthrough infection compared to a reinfection in the naturally immunized [28].

In the present study, there were vaccinated people who were immunologically naïve or who were vaccinated after passing through the infection. Among them, the VE was significantly higher in those that were revaccinated after infection and comparable to that in the naturally immunized (left-censored). Similarly, the cohort study in the general population that was conducted in Qatar between December 2020 and September 2021, 6 months after vaccination with Comirnaty, found a better protection for breakthrough infection in those that were vaccinated after a first episode of COVID-19 compared to those that were vaccinated as immunologically naïve persons, with an adjusted hazard ratio of 0.62 (95% CI: 0.42–0.92) [29].

In addition to the level of VE, the lower number of cases in the waves of the pandemic in 2021, when the largest number of healthcare personnel were immunized through vaccination and revaccination, supports the significant vaccine protection for the prevention of symptomatic cases. Similarly, the better protection for the second episode in those that were vaccinated or revaccinated after a first COVID-19 episode supports vaccine protection.

The advantages of the study are the detailed analysis of the cases, the active and homogeneous surveillance of the sample, the complete participation of the healthcare staff during the study period, the correct classification according to the authentic state of each person, and the correlation with the time of vaccination.

The main disadvantages of the study are the sample size, the lack of representation of the general population, and the possibility of undetected biases despite the active surveillance of the cases.

## 5. Conclusions

The effectiveness of the anti-COVID-19 vaccination for the prevention of symptomatic forms in healthcare personnel that were exposed to the SARS CoV-2 virus was high, but of short duration. Those who were revaccinated had better protection than the vaccinated, but for a period limited to 3 months.

The protection for the prevention of the second symptomatic episode of COVID-19 was better in those that were revaccinated after passing through infection and comparable to the protection of those that were naturally immunized. Compared to vaccination, the protection following natural infection begins to decrease after a longer period, i.e., after about a year.

Of the symptomatic COVID-19 cases, more than a third were breakthrough infections. Their share was lower in the case of the revaccinated compared to the vaccinated persons.

Those with the second symptomatic episode were mainly unvaccinated people, had milder clinical forms, and needed medical care much less often than those with the first episode.

In addition, vaccine protection was shown by the reduced number of cases among healthcare workers in the first 12 months after the introduction of vaccination and the higher share of unvaccinated cases among those with two episodes of COVID-19.

The results of such a VE study in the field are useful for a vaccine policy in guiding evidence-based decision-making to optimize vaccination program performance. In addition, these results are useful for the development of educational programs for the general population to support the immunizations.

## Figures and Tables

**Figure 1 vaccines-11-00521-f001:**
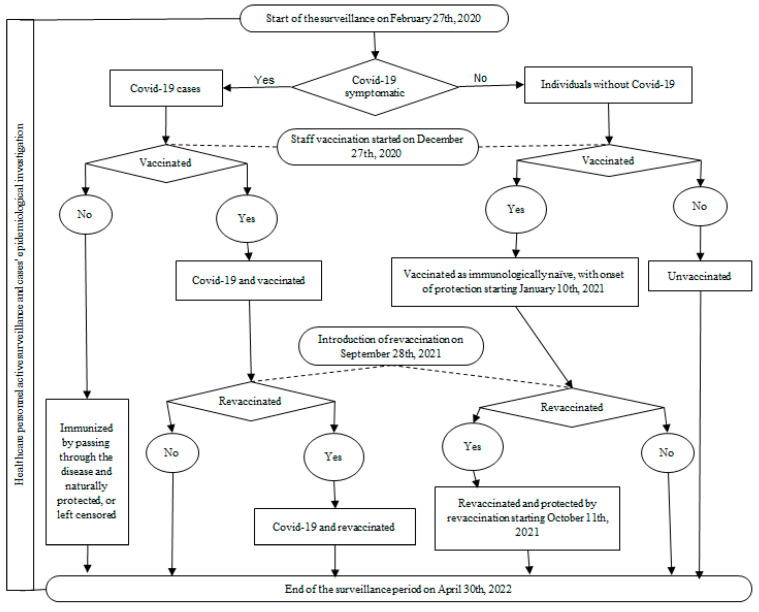
The workflow graphic for the active surveillance of the healthcare personnel and the prospective cohort study for anti-COVID-19 vaccination effectiveness assessment.

**Figure 2 vaccines-11-00521-f002:**
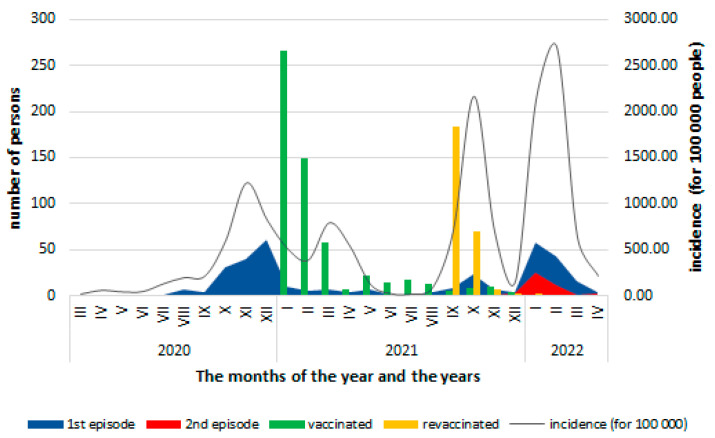
Distribution of COVID-19 cases according to the month in which they were confirmed, the vaccinated persons by the start of vaccination and the revaccination, and the monthly incidence (per 100,000 inhabitants) of COVID-19 at the national level (data source: [4]).

**Figure 3 vaccines-11-00521-f003:**
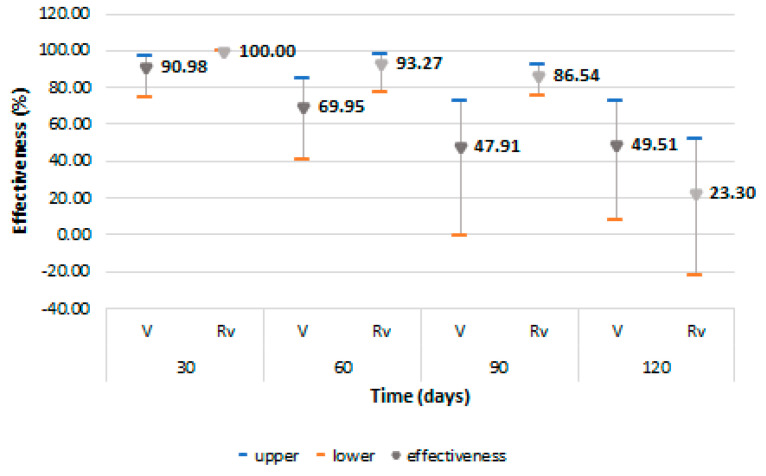
Vaccine effectiveness and 95% confidence intervals for prevention of first COVID-19 symptomatic episode among vaccinated and revaccinated persons. Abreviations: Rv = revaccinated; V = vaccinated.

**Figure 4 vaccines-11-00521-f004:**
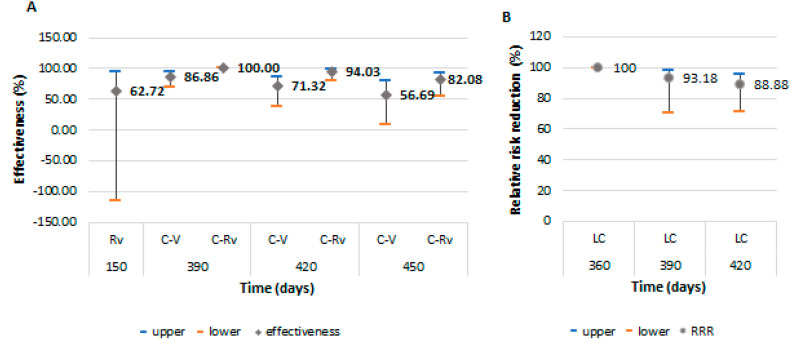
Vaccine effectiveness and 95% confidence intervals for prevention of a second COVID-19 symptomatic episode, and protection from recurrence in those that were naturally immunized after passing through infection. (**A**): the effectiveness among revaccinated persons as immunologically naïve and vaccinated or revaccinated persons after passing through the infection; (**B**): the protection for the second COVID-19 symptomatic episode of persons naturally immunized after infection. Abreviations: C-Rv = COVID-19 and revaccinated; C-V = COVID-19 and vaccinated; LC = left-censored; Rv = revaccinated; V = vaccinated.

**Table 1 vaccines-11-00521-t001:** Main characteristics of the healthcare personnel that were monitored during January 2021–April 2022, according to the presence of the COVID-19 symptomatic episode.

Parameter	Sample (*n* = 783)		First COVID-19 Episode (*n* = 335)		Second COVID-19 Episode (*n* = 40)		Without COVID-19 (*n* = 448)		*p* ^a^
Age (Years)	Value	Range	Value	Range	Value	Range	Value	Range	
Mean	45.06	19.65–68.76	44.17	19.99–68.54	41.85	25.98–59.72	45.72	19.65–68.76	0.054
Gender	no	%	no	%	no	%	no	%	
Female	641	81.86	277	82.69	32	80.00	364	81.25	0.691
Male	142	18.14	58	17.31	8	20.00	84	18.75	0.691
Occupation									
Administrative	116	14.81	50	14.92	10	25.00	66	14.73	0.628
Care staff	193	24.65	64	19.10	4	10.00	129	28.80	0.009
Nurse	291	37.16	145	43.28	22	55.00	146	32.59	0.010
Pharmacist	7	0.89	4	1.20	0	0	3	0.67	0.707 ^b^
Physician	176	22.48	72	21.49	4	10.00	104	23.21	0.311
Clinical unit category									
Administration	115	14.68	52	15.52	10	25.00	63	14.10	0.331
Ambulatory	180	22.98	87	25.97	8	20.00	93	20.76	0.133
Clinical unit	377	48.15	148	44.18	14	35.00	229	51.12	0.025
Pharmacy	8	1.02	5	1.49	0	0	3	0.67	0.480 ^b^
Laboratory	72	9.20	32	9.55	5	12.50	40	8.93	0.719
Radiology	31	3.96	11	3.28	3	7.50	20	4.46	0.726
Vaccination									
Fully vaccinated -before first episode -after first episode	570 463 107	72.80 59.13 13.67	245 138 107	73.13 41.19 31.94	23 3 20	57.50 7.50 50.00	325 - -	72.54 - -	0.033 0.004 0.041
Booster -before first episode -after first episode	262 222 40	33.46 28.35 5.11	101 58 43	30.15 17.31 12.83	7 1 6	17.50 2.50 15.00	161 - -	35.94 - -	0.031 0.011 0.609
Non-vaccinated	213 ^c^	27.20	90 ^d^	26.87	17	42.50	123 ^e^	27.46	0.046
Vaccine type									
Comirnaty	534 ^c^	68.20	225 ^d^	67.16	20	50.00	309 ^e^	68.97	0.251
Janssen	33	4.22	18	5.37	2	5.00	15	3.35	0.330
Vaxzevria	5	0.64	2	0.60	1	2.50	3	0.67	0.197
Spikevax	1	0.13	1	0.30	0	0	0	0	0.377 ^b^

N: the number of persons in the sample; *p*: *p*-value of the statistical significance test. ^a^ *p*-value for the horizontal comparison among the percentages between the categories of groups, the test used was Kruskal–Wallis for mean values and Fisher’s exact test for qualitative variables. ^b^ The *p*-value was calculated with the unit of one patient. ^c^ three people were partially vaccinated with a dose of Comirnaty. ^d^ One person was partially vaccinated, with a dose of Comirnaty. ^e^ Two people were partially vaccinated with a dose of Comirnaty.

**Table 2 vaccines-11-00521-t002:** The clinical and epidemiological descriptions of COVID-19 cases in the first and second episode of the illness among healthcare staff that were monitored during January 2021–April 2022.

Parameter	First COVID-19 Episode (*n* = 335)		Second COVID-19 Episode (*n* = 40)		*p*-Value
	No.	%	No.	%	-
Clinical forms					
Critical disease	1	0.29	0	0	0.976 ^a^
Severe disease	2	0.60	0	0	0.994 ^a^
Moderate disease	58	17.32	3	7.50	<0.0001
Mild disease	274	81.79	37	92.50	<0.0001
Hospitalization					
Intensive care	1	0.30	0	0	0.996 ^a^
Hospital stay/hospitalization	51	15.22	0	0	0.003 ^a^
Outpatient evaluation	193	57.61	11	27.50	<0.001
Without evaluation	90	26.87	29	72.50	<0.0001
Associated diseases					
Present concomitant diseases	145	43.28	18	45.00	0.527
Cardiovascular diseases	66	19.70	10	25.00	0.549
Obesity	47	14.03	7	17.50	0.644
Endocrine and metabolic diseases	40	11.94	3	7.50	0.607
Allergic diseases	33	9.85	6	15.00	0.491
Digestive diseases	18	5.37	1	2.50	0.709
Respiratory diseases	17	4.78	3	7.50	0.475
Neoplasia	5	1.49	1	2.50	0.506
Other diseases	48	14.33	3	7.50	0.341
Source of contamination					
unknown	180	53.73	20	50.00	0.738
community	113	33.73	16	40.00	0.482
patients	25	7.46	1	2.50	0.338
colleague	9	2.69	0	0	0.606 ^a^
office	8	2.39	3	7.50	0.101

no: number; *p*: *p* value of the statistical significance test. ^a^
*p*-value calculated using the unit of a patient instead of zero.

## Data Availability

The primary data can be provided by the corresponding author upon reasonable request.

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
