# Peer review of "Effectiveness and Protection Duration of Anti-COVID-19 Vaccinations among Healthcare Personnel in Cluj-Napoca, Romania"

_vaccines, 2023, doi:10.3390/vaccines11030521_

Round 1

Reviewer 1 Report

The paper analyzed the anti-Covid-19 vaccine effectiveness (VE) and the duration of protection against symptomatic forms of infection, among the healthcare personnel, professionally exposed to SARS-CoV-2 virus in Cluj-Napoca, Romania During January 2021 and April 2022. They find the highest VE in preventing the symptomatic forms of Covid-19 was in the revaccinated, but limited to 3 months duration. The VE for the revaccinated was 93.27% (95% CI: 77.53-97.99) at 60 days and 86.54% (95% CI: 75.59-92.58) at 90 days after revaccination. A better protection against reinfection was provided by revaccination after passing through infection. Protection against reinfection was 94.03% (95% CI: 79.41-98.27) at 420 days and 82.08% (95% CI: 53.93 -93.03) at 450 days after revaccination for previously infected personnel. Due to case number limitation (only one case in this category), the effectiveness for revaccinated couldn’t be calculated.

only minor change:

line 19: "profesionally"should be:"professionally"

Author Response

Point 1: ine 19: "profesionally"should be:"professionally"

Response 1: Thank you for the recommendation. We corrected in the text the word “professionally” from line 19 and the correction is tracked in the revised manuscript.

Reviewer 2 Report

Thank you for giving me this opportunity to review this valuable manuscript. The authors conducted a prospective study with active surveillance using RT-PCR to evaluate the effectiveness of vaccination against Covid-19. They gathered information on vaccination status, and characteristic study participants including age, sex, occupation, and unit. The study objective is reasonable, however, there are some points to be considered.

 1.    The study design and statistical analysis were not clear. The readers can’t reproduce the results in Supplemental Tables and Figure 2, because of the exact numbers of cases in vaccinated and unvaccinated. If this study was prospective cohort study, they could compare the incidence of infection and evaluate hazard ratios and their 95 % confidence intervals using Cox regression.

2.    Though the study population was limited the healthcare works, there were several significant differences between participants with and without COVID-19 episode(Table1), those might be confounded between vaccination and CPVID-19 episodes. It had better check whether those factors had confounding, they had better adjust for confounding factors using multivariable hazard ratio analysis. 

3.     Because the study period was long, there were three epidemic waves as they showed in Figure 1. Though they did not examine viral lineage, they should be adjusted for the study period. 

4.    Abstract: They should include the number of participants, and the place where the study was conducted. It was not clear whether they evaluated vaccine effectiveness for symptomatic infection or not. 

5.    I recommend the authors fill out the STROBE checklist to make sure they have covered all essential items accordingly.

Author Response

Response to Reviewer 2 Comments

Point 1: The study design and statistical analysis were not clear. The readers can’t reproduce the results in Supplemental Tables and Figure 2, because of the exact numbers of cases in vaccinated and unvaccinated. If this study was prospective cohort study, they could compare the incidence of infection and evaluate hazard ratios and their 95 % confidence intervals using Cox regression.

Response 1: Thank you for these remarks.

For making more clear the study design, we have added a workflow grapfic as figure 1 (see below), to the chapter Material and method. Thus the order number of the figures has been changed. We inserted the figure in the manuscript.  

Figure 1. The workflow graphic for the active surveillance of the healthcare personnel and the prospective cohort study for anti-Covid-19 vaccination effectiveness assessment.

To improve the figure 2 (former), we have splitted it into two figures. The derived figures are the current figure 3 (see below), for the effectiveness in preventing the first Covid-19 episod and the figure 4 (see below), with the effectivenes for preventing a second symptomatic episode and the protection of those who were immunized after passing through the infection. Both figures, 3 and 4, were also inserted in the manuscript.    

Figure 3. Vaccine effectiveness and 95% confidence intervals for prevention of first Covid-19 symptomatic episode among vaccinated and revaccinated persons. Abreviations: Rv=Revaccinated; V=vaccinated.

Figure 4. Vaccine effectiveness and 95% confidence intervals for prevention of a second Covid-19 symptomatic episode, and protection from recurrence in those naturally immunized after passing through infection. A: the effectiveness among revaccinated as immunologically naive and vaccinated or revaccinated after passing through infection; B: the protection for the second Covid-19 symptomatic episode of persons naturally immunized after infection. Abreviations: C-Rv=Covid-19 and revaccinated; C-V=Covid-19 and vaccinated; LC=Left censored; Rv=Revaccinated; V=vaccinated.

We are aware that the Cox model is better to measure vaccine effectiveness. In our study the main criteria for its use have not been met. For the majority of covariate the p was smaller than 0.05 and the global value of p was <0.0001. This indicating a violation of the proportionality assumption.

   The supplementary data, indeed they provide too little information and we will remove them.  

Point 2: Though the study population was limited the healthcare works, there were several significant differences between participants with and without COVID-19 episode(Table1), those might be confounded between vaccination and CPVID-19 episodes. It had better check whether those factors had confounding, they had better adjust for confounding factors using multivariable hazard ratio analysis.

Response 2: Thank you for this observation. The only possible confounding factor we suspected was the gender, but for which no significant difference was found in the medical literature. We included in the Discussion section, lines 279-281, the results from the literature:

“Among the healthcare personnel, the female predominance had no influence on the VE, the studies carried out so far showing that the difference between the sexes regarding the immune response to the Covid-19 vaccines was statistically insignificant [16].”

Point 3: Because the study period was long, there were three epidemic waves as they showed in Figure 1. Though they did not examine viral lineage, they should be adjusted for the study period.

Response 3: Thank you for this observation. Regarding the circulating variants, we presented in the discussion part, lignes 317-321 (“The predominant variants circulating nationally in the last months of 2020 were the ancestral strain, the Alpha strain in 2021, in the first 6 months, then in the second half, Delta, followed by Omicron in the first months of 2022 [25]. In the present study, the circulating variants were not followed because genomic sequencing was not performed on staff cases.”), the circulating variants at national level, but in our study there weren’t made genomic sequencings for our cases and also we didn’t plan to analyze the correlation of protection with the SARS-CoV-2 strains.

Point 4: Abstract: They should include the number of participants, and the place where the study was conducted. It was not clear whether they evaluated vaccine effectiveness for symptomatic infection or not. 

Response 4: Thank you for this remark. The reason for not including in the abstract some general information was the limitation to a certain number of words to be accepted. However, we also consider important the sample size and therefore, we added in the abstract the number of persons included in the study sample, in the results part: “Among the 783 subjects included in the study...”.

Regarding the location, we added in the abstract that the study was conducted in a „university hospital”, however, we could’t mention the hospital’s name due to privacy precautions, and to avoid any possible discrimination of the healthcare personnel.    

Point 5: I recommend the authors fill out the STROBE checklist to make sure they have covered all essential items accordingly.

Response 5: Thank you for the recommendation. After we verified the STROBE Statement—Checklist of items that should be included in reports of cohort studies, for the item 5, the location we like to not write exactly the name of hospital for privacy reasons and for the item 12, for confounding, a possible confounding factor we suspected the gender for which in the literature we found a systematic revue (index 16 in References) which concludes that there is no significant difference (please see above response to Point 2).  

Reviewer 3 Report

Dear Editor,

I am sharing my review of the Manuscript ID vaccines-2214303 entitled: Effectiveness of the anti-Covid-19 vaccination and the duration of protection among healthcare personnel, in Cluj-Napoca, Romania.

This study aimed to measure the anti-Covid-19 vaccine effectiveness (VE) and the duration of protection against symptomatic forms of infection among the healthcare personnel exposed to SARS-CoV-2 virus. A prospective cohort study was conducted between January 2021 and 20 April 2022 and included vaccinated and revaccinated, either immunologically naive or previously infected, and unvaccinated personnel. The highest VE in preventing the symptomatic forms of Covid-19 was in the revaccinated but limited to 3 months. Better protection against reinfection was provided by revaccination after passing through infection.

MM section:

A workflow graphic overview would be helpful.

Result section:

Figure 1 is a bit blurry and should be improved.

The manuscript should be accepted after minor revisions.

Author Response

Response to Reviewer 3 Comments

Point 1: MM section:

A workflow graphic overview would be helpful.

Response 1: Thank you for this recommendation. We have structured a workflow chart representing the working method in the study (see below) and we included it in the manuscript as figure 1. Thereforre we changed the order number of figure and especially of the following two graphic figures 2 and 3.  

Figure 1. The workflow graphic for the active surveillance of the healthcare personnel and the  prospective cohort study for anti-Covid-19 vaccination effectiveness assessment.

Point 2: Result section:

Figure 1 is a bit blurry and should be improved.

Response 2: Thank you for this recommendation. The graph in figure 1 (former), aims to document the presence of a high exposure to the SARS-CoV-2 virus in the community as an important condition for measuring of the vaccination program effectiveness through a cohort study. Taking this into account and to make the graph more clearly we accentuated the colors, added a title to the horizontal axis and bolded some words. This modified graph has been inserted in the manuscript visible with the tracking change.

We also reproduce it here below.  

Figure 2. Distribution of Covid-19 cases according to the month in which they were confirmed, the vaccinated persons by the start of vaccination and the revaccination, and the monthly incidence (per 100,000 inhabitants) of Covid-19 at the national level (data source: [4]).

Round 2

Reviewer 2 Report

Thank you for your revision according to this reviewer's comment. I confirmed them corrected well, and find I have no further comment.